# The Role of Zinc in the Treatment of Wilson’s Disease

**DOI:** 10.3390/ijms23169316

**Published:** 2022-08-18

**Authors:** Abolfazl Avan, Anna Członkowska, Susan Gaskin, Alberto Granzotto, Stefano L. Sensi, Tjaard U. Hoogenraad

**Affiliations:** 1Department of Public Health, School of Medicine, Mashhad University of Medical Sciences, Mashhad 93518-88415, Iran; 22nd Department of Neurology, Institute of Psychiatry and Neurology, 02-957 Warsaw, Poland; 3Department of Civil Engineering, McGill University, Montreal, QC H3A 0C3, Canada; 4Center for Advanced Studies and Technology (CAST), University “G. d’Annunzio” of Chieti-Pescara, 66100 Chieti, Italy; 5Department of Neuroscience, Imaging, and Clinical Sciences (DNISC), University “G. d’Annunzio” of Chieti-Pescara, 66100 Chieti, Italy; 6Sue and Bill Gross Stem Cell Research Center, University of California-Irvine, Irvine, CA 92697, USA; 7Institute for Advanced Biomedical Technologies (ITAB), University “G. d’Annunzio” of Chieti-Pescara, 66100 Chieti, Italy; 8Department of Neurology, University Medical Centre Utrecht, 3584 CX Utrecht, The Netherlands

**Keywords:** Wilson’s disease, copper metabolism, copper intoxication, zinc therapy, chelating agents

## Abstract

Wilson’s disease (WD) is a hereditary disorder of copper metabolism, producing abnormally high levels of non-ceruloplasmin-bound copper, the determinant of the pathogenic process causing brain and hepatic damage and dysfunction. Although the disease is invariably fatal without medication, it is treatable and many of its adverse effects are reversible. Diagnosis is difficult due to the large range and severity of symptoms. A high index of suspicion is required as patients may have only a few of the many possible biomarkers. The genetic prevalence of *ATP7B* variants indicates higher rates in the population than are currently diagnosed. Treatments have evolved from chelators that reduce stored copper to zinc, which reduces the toxic levels of circulating non-ceruloplasmin-bound copper. Zinc induces intestinal metallothionein, which blocks copper absorption and increases excretion in the stools, resulting in an improvement in symptoms. Two meta-analyses and several large retrospective studies indicate that zinc is equally effective as chelators for the treatment of WD, with the advantages of a very low level of toxicity and only the minor side effect of gastric disturbance. Zinc is recommended as a first-line treatment for neurological presentations and is gaining acceptance for hepatic presentations. It is universally recommended for lifelong maintenance therapy and for presymptomatic WD.

## 1. Introduction

Wilson’s disease (WD) is a rare hereditary impairment of copper metabolism with impaired incorporation of copper into ceruloplasmin, leading to free copper intoxication (oxidative stress) and its deposition, mainly in the liver and brain, with lesser amounts in other organs. Disease results due to the toxicity of the deposited copper and/or the oxidative stress from reactive oxygen species (ROS) formed by the non-ceruloplasmin-bound copper [1,2,3]. Presentation is initially recognized due to hepatic (30–50%), neurologic (30–40%), or psychiatric/behavioral (30–40%) disturbances [4], although ophthalmological, renal, cardiac, osteoarticular, hematologic, endocrine, and sleep disturbances can also occur [5,6,7]. The disease presents predominantly between the ages of 5 and 35, although it has also been reported between the ages of 2 and 80 [8]. The large variation in the range, severity, and age of onset of symptoms explains the diverse manifestations of WD; no two cases are the same even in a sibship [9,10]. Patients can also be asymptomatic (or pre-symptomatic) when diagnosed through family screening [4]. Diagnosis should be considered in any patient with unexplained hepatic, neurologic, and psychiatric abnormalities [11]. The disease is fatal, but if diagnosed, safe effective treatments are available, enabling a close to normal lifespan with almost full quality of life [4].

WD is an autosomal recessive disorder of copper metabolism caused by the *ATP7B* gene variants, leading to impaired incorporation of copper into ceruloplasmin in the trans-Golgi network and impaired biliary excretion of excess copper. Over 800 variants have been identified as disease-causing as of 2021 [8,12]. In a study of 1172 WD index patients, no correlation with disease phenotype was identified, of which 40% were heterozygotes and 39% were compound heterozygotes, while in 21%, only one mutation was found [8]. Incidence is usually quoted as 1:30,000 [13,14], but recent genetic studies have shown a higher prevalence, ranging from 1:2400 and 1:3150 to 1:6500 to 1:20,000 if all, only pathogenic, and suspected low penetrance of variants are included, respectively [12,15,16]. The discrepancy between the diagnostic rates and hypothesized genetic prevalence could be attributed to either a combination of low penetrance and/or low rates of diagnosis.

We conducted a critical review of the evidence of the potential mechanisms of current therapeutic options in WD and evaluated the scientific rationale for zinc-based therapy. To accomplish our aims, we also summarized the history and mechanisms of copper toxicity in the etiology of WD and reviewed suggested treatment options.

## 2. Chronological Overview of Copper Disorder and WD

In 1912, the neurologist Kinnier Wilson described the clinical features of four patients with progressive lenticular degeneration and liver cirrhosis [17]. An increased copper concentration in the brain and liver of patients with WD was observed from the early 1930s and confirmed by Cumings in 1948, who suggested that the accumulation of copper was the probable cause of disease [18]. In 1951, Porter reported high urinary copper levels [19] and two groups reported low ceruloplasmin levels in patients with WD [20,21].

Denny-Brown and Porter suggested the use of the chelator British anti-Lewisite (BAL), also known as dimercaprol or dithiopropanol, to mobilize and remove accumulated copper in WD [22]. In 1956, Walshe introduced penicillamine, based on a chemical analysis, as a less toxic chelator able to be taken orally [23], which was shown to increase copper excretion compared to BAL [3]. In 1969, as a treatment of last resort for an individual patient, based on animal studies, Walshe introduced trientine as a less toxic alternative to penicillamine [24], which was later compared to penicillamine [25] and assessed over the longer term [26]. These three chelators mobilize body copper stores and increase the urinary excretion of copper, allowing WD to be treated in an increasing number of patients [27]. However, a large percentage of patients, which is unpredictable at the individual level, either do not improve or deteriorate (e.g., under trientine treatment after withdrawal from D-penicillamine treatment, 30% of hepatic and 90% of neurologic patients either did not improve or deteriorated [28]). Puzzlingly, Scheinberg and Sternlieb, in 1984, reported a ‘paradoxical’ deterioration (clinical worsening with an increase in copper excretion, where increased copper excretion is regarded as a marker of treatment effectiveness) in many patients taking penicillamine [14]. In 1986, Scheinberg and Walshe proposed tetrathiomolybdate treatment for a patient as an alternative chelator [29]. By 1996, Brewer combined zinc salts with 8 weeks of tetrathiomolybdate followed by zinc therapy, with good results observed at the 5-year follow-up in 40 neurologically presenting patients [30]. Bis-choline tetrathiomolybdate (ALXN1840) is a novel oral agent that forms a stable tripartite complex with copper and albumin, causing rapid copper control and having a favorable safety profile, as shown in a stage 2 trial [31]; it is currently undergoing a stage 3 trial [32]. In 1982, liver transplantation was first reported as a treatment of WD [33] and also recommended to treat copper dyshomeostasis in the brain [34]. Plasmapheresis was first used in 1989 to reduce the non-ceruloplasmin-bound copper in two patients, resulting in improved neurologic symptoms [35]. Since then, it has been used to avoid the need for liver transplants in cases of acute liver failure [36].

An alternative approach to chelation was presented in 1961, when the neurologist Schouwink published in his thesis (in Dutch) [37] a comparative pilot study of the effect of penicillamine to zinc sulfate on copper metabolism in two patients, demonstrating that zinc blocks the absorption of copper in the gut and increases fecal copper excretion in stools [37]. In 1978, Hoogenraad et al. reported the first use of zinc treatment for a patient [38], and by 1987, a review of 27 patients, half with zinc monotherapy [39]. Their assessment was that zinc treatment was safe and effective as a first-line treatment even in severe cases, with only occasional temporary gastric upset [37]. Independently, Brewer and colleagues, on the basis of copper deficiency due to zinc treatment of sickle cell anemia, initiated zinc acetate treatment of five WD patients, which proved effective [40]. Brewer subsequently reported on, in a series of 18 papers, comparative studies of zinc treatment [41,42,43,44,45,46,47,48,49,50,51,52,53,54,55,56,57,58], demonstrating its safety and effectiveness. Reviews of the historical development of the treatment of WD have been compiled [59,60,61].

## 3. Mechanisms of Copper Toxicity

Copper-driven cytotoxicity has been primarily linked to the redox properties of the cation. Copper dysregulation participates in Haber–Weiss or Fenton-like reactions, resulting in the aberrant generation of reactive oxygen species [62]. The accumulation of oxidative stress promotes lipid peroxidation, macromolecule damage, and, eventually, cell death [63]. Along with this unspecific modality of cellular demise, the metal can also target and impair specific subcellular compartments.

For instance, copper accumulation has been reported in lysosomes [64]. Although the contribution of copper in lysosomal function/dysfunction has been poorly explored, two main mechanisms have been proposed. On the one hand, lysosomes act as copper-buffering organelles and defective metal uptake results in the toxic build-up of the cation in the cytosol and other subcellular compartments. Alternatively, reduced copper import into lysosomes might impair the correct functioning of copper-dependent enzymes of the organelles, hence resulting in the accumulation of dysfunctional, partially metabolized molecular byproducts [64].

Mitochondria require copper as a cofactor for proper assembly of the protein complexes of the electron transport chain. However, when dysregulated, copper targets the organelles and alters their functioning [65]. In this regard, it has recently been proposed that increased copper accumulation triggers a mitochondrial-dependent form of cell demise that differs from other subtypes of cell death (i.e., apoptosis, necrosis, and ferroptosis) and, hence, is termed “cuproptosis”. Cuproptosis requires several molecular intermediates to occur, such as ferredoxin 1 (FDX1), an iron-dependent effector of protein lipoylation, a mechanism that results in toxic aggregation of key metabolic enzymes [66,67]. Although it is still unclear whether cuproptosis is involved in WD, this study identified novel and potentially relevant therapeutic targets. In addition, the work pinpoints a deadly liaison between copper and iron-containing molecules, suggestive of a possible interplay among metals in cell death [68].

Along with the proposed role in cytotoxic processes, copper dysregulation can also impair proper neurotransmission, a mechanism that might be relevant for the neurological manifestations of WD or, as it has been extensively indicated, some subsets of patients affected by Alzheimer’s disease [69]. The metal modulates synaptic transmission as the cation is released from copper-rich vesicles [70]. Post-synaptically, it has been found that copper can, directly or indirectly, modulate the activity of several neurotransmitter receptors, such as NMDA, AMPA, and GABA receptors [71], thereby providing compelling evidence for copper-mediated regulation of neuronal excitability.

## 4. Different Mechanisms Postulated to Underlie Treatment

The natural history of WD without treatment is usually progression of either or both liver disease and neurologic disease, and eventually death. WD is, however, one of the few treatable metabolic diseases. The two approaches to the treatment of WD, chelation and reduction of serum non-ceruloplasmin-bound copper levels, are based on the two premises of the cause of symptoms. The first cause is due to the storage of excess copper in organs, primarily the liver and the brain, and the second is due to oxidative stress arising from the high levels of circulating non-ceruloplasmin-bound copper. Normal and impaired copper metabolism are illustrated in Figure 1a,b, respectively.

### 4.1. Copper Accumulation in Organs

Copper accumulates in the liver and brain, and in other organs, due to greater ingestion than excretion of copper in WD. Excess copper is at first stored in the liver, whose damage presents as hepatitis, liver failure, or chronic cirrhosis [4]. With increased levels of circulating non-ceruloplasmin-bound copper, copper is deposited in other organs. The next most sensitive organ is the brain, in which copper accumulates as non-ceruloplasmin-bound copper, which can cross the blood-brain barrier [76,77,78]. Increased copper levels have also been observed in other organs, including the kidneys, eyes, heart, bones, and other organs, resulting in a range of clinical manifestations [5].

The chelators penicillamine and trientine are recommended as treatments as they lower body copper levels by mobilizing stored copper as non-ceruloplasmin-bound copper for urinary excretion [4,79], as shown in Figure 1c. Urinary copper levels are increased in the de-coppering phase of treatment (more for penicillamine than trientine) and then level off to roughly normal values with concomitant improvement in the liver function test for the majority of hepatically and neurologically presenting patients [80]. However, the mobilized copper, circulating as non-ceruloplasmin-bound copper, crosses the blood-brain barrier, resulting in a lack of neurologic improvement or neurologic worsening in patients, half of whom remain permanently disabled, under penicillamine (50% of patients) and trientine (15%) treatment [81,82]. These percentages decrease when a gradual increase in the chelator dose modulated to the response is administered [83]. In addition, penicillamine and trientine have many serious adverse effects, occurring in 25–30% of patients [28,84]. Recently, exchangeable copper levels have been proposed to differentiate between hepatic and extrahepatic presentations of WD, thereby identifying those more likely to suffer from neurologic worsening with chelation treatment [85].

### 4.2. Free Copper Intoxication/Oxidative Stress

Non-ceruloplasmin-bound or free copper is the damage-causing moiety of copper [1,14,61,82] as it results in oxidative stress [62]. Neurologic worsening with the use of chelators occurs due to increases in non-ceruloplasmin-bound copper levels [81].

Zinc salts (sulfate, gluconate, acetate) act to reduce non-ceruloplasmin-bound copper levels, resulting in improvements in neurologic symptoms, but must be taken away from food in divided doses over the day [1,52]. Zinc salts induce intestinal metallothioneins, which safely bind copper in food and endogenous secretions to be sloughed off and excreted in the stools. Zinc also induces metallothioneins in hepatocytes, lowering non-ceruloplasmin-bound copper circulating in the blood and resulting in a gradual de-coppering of organs [1,58]. Non-ceruloplasmin bound copper levels can be estimated by subtracting the copper bound to ceruloplasmin from the total serum copper [86]. This is accurately estimated if the holo-ceruloplasmin levels are determined enzymatically; however, it may be underestimated (but never over-estimated [4]) if the clinically available immunological method is used (which measures both holo- and apo-ceruloplasmin, the latter with no bound copper) [87,88]. Alternatively, exchangeable copper, a process-based measurement of loosely bound copper, can be monitored as it reflects the severity of neurological symptoms [85].

## 5. The Challenge of Diagnosis

Diagnosis of WD is challenging due to the range of symptoms and their severity; no two cases are the same [8]. Hepatic presentation ranges from acute presentations (acute hepatitis, fulminant liver failure) to chronic presentations (steatosis, chronic hepatitis, compensated and decompensated cirrhosis) and hemolysis [4,79]. Neurologic symptoms are typically movement disorders (tremor, dystonia, parkinsonism), bulbar symptoms (dysarthria, drooling, dysphagia), and other symptoms (including cerebellar dysfunction, chorea, hyperreflexia, seizures, cognitive impairment) [11]. Psychiatric symptoms are often apparent due to a decrease in scholastic or work performance and include behavioral problems, affective disorders, and psychosis [11].

Diagnosis of WD is based on clinical presentation and a number of biomarkers, of which the majority are measurements related to copper metabolism or liver function [4,79]. None of the current tests are specific or sensitive on their own [4,89] and by corollary, no single test can be used to exclude the diagnosis. Clinical presentation can be assessed systematically using the Unified Wilson’s Disease Rating Scale [90,91] or the Global Assessment Scale for WD (GAS) [92]. Several composite diagnostic rating scales have been developed, the original and most commonly used being the Leipzig score, which assesses seven symptoms or biomarkers resulting, on the basis of one to four indicators, in an evaluation of a diagnosis being established (four points), possible (three points), or very unlikely (two points) [4]. The European Association for Study of Liver (EASL) guidelines [4] present, in addition, five routine tests (serum ceruloplasmin, 24-h urinary copper, serum free copper, hepatic copper, and presence of Kayser–Fleischer rings with explanations of the occurrence of false negative and false positive results). Family screening is required following diagnosis as WD is an autosomal recessive disorder [4].

As WD may present as only hepatic or only neurologic, this reduces the number of biomarkers that could be relevant for an individual case. There is no gold standard as all biomarkers can result in false negatives if they are not present, and no biomarkers can exclude diagnosis. Serum ceruloplasmin can be elevated with inflammation, increase estrogen, and be over-estimated by immunologic assay; it has a 6% predictive value [4]. Urine copper can be low in children without liver disease [4]. Non-ceruloplasmin-bound copper can be a false negative (but never a false positive) when ceruloplasmin is measured immunologically [88] or when ceruloplasmin is elevated. The hepatic copper distribution is very heterogeneous, and its concentrations can be below the cut-off in a high percentage of WD patients [93]. Kaiser–Fleischer rings are absent in up to half of hepatic presentations and 5–10% of neurologic presentations [4]. Detailed genetic analyses indicate that 20% of WD patients have only one *ATP7B* variant [8] while standard genetic testing only tests for the most common variants, resulting in confirmation of only 50–75% of cases [74]. The relative frequency of the diagnostic markers determined in two retrospective studies is provided in Table 1, which indicates that the probability of a patient having any three of these biomarkers is between 50 and 70% [89].

Copper toxicosis is due to high levels of non-ceruloplasmin-bound copper, which has long been used to monitor treatment efficacy [4,79] and is proposed for diagnosis [1,4]. However, the calculated non-ceruloplasmin-bound copper available clinically can result in false negative values [4] when ceruloplasmin levels are high (high proportion of apo-ceruloplasmin compared to the active holo-ceruloplasmin form [88]), requiring repeated measurements over several months. A recently developed biomarker providing a direct measure of loosely bound (or non-ceruloplasmin-bound copper) is exchangeable copper, a process-based measure [96]. Relative exchangeable copper, which is the ratio of the exchangeable copper to the total serum copper, has been shown to have high sensitivity and specificity in the diagnosis of WD [96,97].

New biomarkers are being developed to aid in diagnosis, with a particular motivation being the identification of those who will suffer from permanent neurological worsening under chelation therapy [98]. Neuroimaging methods and biomarkers derived therefrom are being explored [78] and neurofilament light concentrations in plasma have been found to correlate with neurologic damage [98]. A direct measurement of ATP7B peptides in dried blood spots is showing promise as a diagnostic screening tool [99].

## 6. Effectiveness and Safety Profile of Treatment Options

There have been no randomized controlled trials (RCTs) of monotherapies in WD, and therefore the results of concomitant therapy for the disease cannot be used to establish the effectiveness of zinc or chelator monotherapy. The one RCT compared trientine and zinc to tetrathiomolybdate and zinc [100]; however, it was performed on a small group of patients. There are now efforts to design clinical trials in WD to compare the efficacy and safety of different treatments [101]. Nevertheless, two recent systematic reviews and meta-analyses sought to analyze available evidence on the improvement rate and safety profile of chelators and zinc salt treatment in WD patients [102,103]. The pooled analysis of data from 16 studies showed a comparable efficacy of both options for hepatic WD patients but higher efficacy of zinc over D-penicillamine for neurological WD patients [102], noting that the side effects with zinc are transient gastric discomfort while those from D-penicillamine are more severe or permanent. Further, the results proved the higher incidence of both adverse events and neurological deterioration in those treated with D-penicillamine compared to those treated with zinc salts [102]. The analysis of data from 23 studies was also suggestive of a comparable efficacy and again a far lower toxicity of zinc over penicillamine in preventing or reducing hepatic or neurological WD symptoms [103]. One ongoing RCT is comparing tetrathiomolybdate vs. chelation vs. zinc vs. combined chelation and zinc [104].

On the basis of a lack of a well-designed comparative RCTs, we critically analyzed the evidence from clinical experience. The use of chelators resulted in WD changing from being universally fatal to being a treatable disease [105,106,107]. A small number of case series support the use of penicillamine to treat severe hepatic insufficiency [108]. However, a high percentage of patients, who cannot be identified pre-treatment, suffer from ‘paradoxical’ deterioration as chelators promote a transient increase in free copper intoxication [2,109,110] (Figure 1c). The brain is more sensitive to increased levels of free copper [1,14,111]. Moreover, several studies, including some long-term investigations, indicate that treatment with penicillamine, trientine, or tetrathiomolybdate may be accompanied by serious adverse effects [4,31,111,112,113,114,115,116,117,118,119,120,121], some of which are irreversible and fatal [106,115,122]. These drugs are also expensive for patients and health services [123,124].

In a study of 467 European patients [125], chelating agents (i.e., penicillamine or trientine) had better success with hepatic than neurologic WD. In first-line and second-line treatment of hepatic patients, worsening occurred in 10% and 30% of patients, respectively, while for neurologic patients, more than one-third showed non-improvement or worsening, with trientine being worse than penicillamine [125]; however, worsening rates have decreased with the gradual introduction of chelation treatment. Other observational studies have reported that up to 50% of patients with neurologic symptoms experienced deterioration with chelators [81,114,126,127] while their withdrawal was associated with clinical improvement [127].

In two comparative non-randomized cohort studies, zinc and penicillamine, given as initial treatment, were considered equally effective and therefore alternatives for first-line therapy in both hepatic and neurologic WD patients [83,128]. Zinc was, therefore, recommended for patients with WD, particularly for neurologic and presymptomatic patients, due to lower rates of neurologic worsening, fewer side effects, and better compliance [83,128]. No suggestions were made about the treatment of choice for patients with severe hepatic dysfunction because few such patients were included in this study. Small and large case series have suggested that zinc therapy is safe for long-term initial and maintenance treatment of presymptomatic WD in children, pregnant women, and patients with neurological manifestations [53,57,129,130].

Differing from the action of chelators, zinc—in either sulfate, gluconate, or acetate forms—reduces the intestinal absorption of copper, thereby leading to the normalization of free copper levels, depletion of stored copper, and appreciable clinical improvement [37,130,131]. However, treatment compliance is a challenge, as for all WD treatments [132], and non-compliance results in deterioration [133]. Although a few studies have failed to identify zinc therapy as being effective in hepatic impairment of WD patients [107], these studies were retrospective, and no rigorous dose–response or adherence studies are available. Studies have demonstrated that the putative toxicities of pancreatitis, adverse effects on lymphocytes, and adverse effects on cholesterol metabolism do not occur with zinc treatment [45,49,54,55]. In our experience, zinc therapy can be effective for patients with moderate/severe hepatic damage or even fulminant hepatic failure and decompensation, which is supported by reports in the literature [83,134,135,136]. The limited number of patients with severe hepatic damages or acute liver failure treated with zinc might be due to current recommendations that do not suggest the use of zinc as an initial treatment option [4,74,79]. Plasmapheresis or plasma exchange can be used to initially rapidly reduce non-ceruloplasmin-bound copper levels [36] before the initiation of treatment. Normalization of liver function tests should not be regarded as the goal of treatment, rather clinical improvement, which is usually achieved after normalization of free copper levels while gradual improvement of liver function will follow. There are reports of deterioration of symptoms generated by zinc therapy, although in advanced cases, it could be indistinguishable from the natural course of the disease [137,138]. The most common side effect of zinc therapy is gastric upset [139,140], a temporary discomfort that decreases as symptoms improve. Zinc has an otherwise excellent safety profile [141].

## 7. Discussion

The optimal goal for treatment of patients with WD should be the normalization of levels of free copper. Chelators initially increase these levels by mobilizing accumulated copper, as proven by the penicillamine challenge test [142,143]. A decrease in free copper is usually achieved with chelators after 1–2 years [142,144]. In contrast, the induction of metallothionein and the reduction of non-ceruloplasmin-bound copper levels can be persistently achieved from the very beginning of zinc therapy [61,75,145]. The finding that zinc increases copper excretion in feces and reduces copper excretion in urine indirectly confirms that zinc normalizes serum levels of free copper [37].

Treatment guidelines have evolved with clinical experience and research. The original practice guidelines by the American Association for Study of Liver Diseases (AASLD) in 2008 [79] and by EASL in 2012 [4] recommended initial treatment of symptomatic patients with D-penicillamine or trientine while EASL [4] recommended zinc as first-line therapy for neurological presentations. Both recommended lower doses of chelators or zinc for presymptomatic patients and maintenance therapy. New national guidelines have been published in India [146], which also recommends zinc as an alternative first-line therapy for all presentations and as the choice for pre-symptomatic patients. The new guide in the UK has a detailed description of the range of clinical presentations; however, it recommends chelators as first-line therapy due to a lack of experience with zinc salts [147]. The Hepatology Committee of the European Society for Paediatric Gastroenterology, Hepatology and Nutrition’s recommendation for pediatric patients is also similar [148]. All state that treatment is lifelong and monitoring is necessary [4,79,146,147,148,149].

The clinical practice of individual groups has also evolved on the basis of their experience. In the Netherlands, Hoogenraad recommended first-line zinc therapy for all presentations due to the great but unpredictable danger of neurological worsening [1]. In the USA, Brewer recommends zinc therapy as the first choice for all presentations (initial hepatic without decompensation, neurologic/psychiatric, presymptomatic, pediatric, and pregnant), except for hepatic decompensation for which combined trientine and zinc is recommended [74,84]. In Poland, a retrospective study indicated that zinc could be used as a first-line therapy with equivalent results to penicillamine in both hepatic and neurologic presentations [83]. Zinc treatment was approved by the Food and Drug Administration (in the USA) in 1997 and subsequently in Europe and Japan [84,150]. Clinicians may face challenges in choosing zinc as the initial treatment option, due to the relative paucity of literature reporting on its use [124]. 

Legislation in the United States and Europe Union encourages the pharmaceutical industry to develop drugs for rare life-threatening diseases, although these are often extremely expensive [151]. The annual cost of lifetime oral treatment of WD depends on the treatment modality: it ranges from less than 100 US dollars (nutriment) to 550 to 3650 US dollars (pharmaceutical grade) for zinc salts, from 78,000 US dollars or 180,000 Euros for trientine, and from 50,000 to 200,000 US dollars for penicillamine. The cost of a liver transplant, the most radical therapy, is estimated at about 600,000 US dollars, a price that does not include the expensive but mandatory preoperative evaluations (costing 168,000 to 308,000 US dollars) and postoperative medications (costing 671,000 to 949,000 US dollars in the 10 years that follow the procedure) [152]. Any of these treatments can produce results that range from complete clinical improvement and low disease burden to ineffective outcomes, leading to irreversible side effects or even death [122]. 

## 8. Conclusions

Wilson’s disease (WD) is a rare hereditary impairment of copper metabolism with impaired incorporation of copper into ceruloplasmin due to mutations in *ATP7B*, leading to free copper intoxication (oxidative stress) and deposition of copper, mainly in the liver and the brain, with lesser amounts in other organs. Since its identification in 1912, WD has changed from being universally fatal to being treatable with lifelong therapy and monitoring, although the outcome is dependent on the treatment agent and adherence to therapy. Diagnosis is challenging due to the large range and severity of symptoms. Recent progress includes a relative measure of non-ceruloplasmin-bound copper, relative exchangeable copper, which has high specificity and sensitivity. Other new biomarkers are under development. Treatments were initially developed to reduce the organ copper load, with penicillamine and trientine resulting in an improvement in symptoms in the majority of patients despite adverse effects, with better outcomes for hepatic than neurologic/psychiatric presentation. As an alternative to chelation, zinc therapy aims to lower circulating non-ceruloplasmin-bound copper levels. Zinc is as effective as other anticopper agents and has the advantage of an extremely low level of toxicity and minimal adverse effects as shown by its long, effective, and safe use (at an affordable cost). 

## Figures and Tables

**Figure 1 ijms-23-09316-f001:**
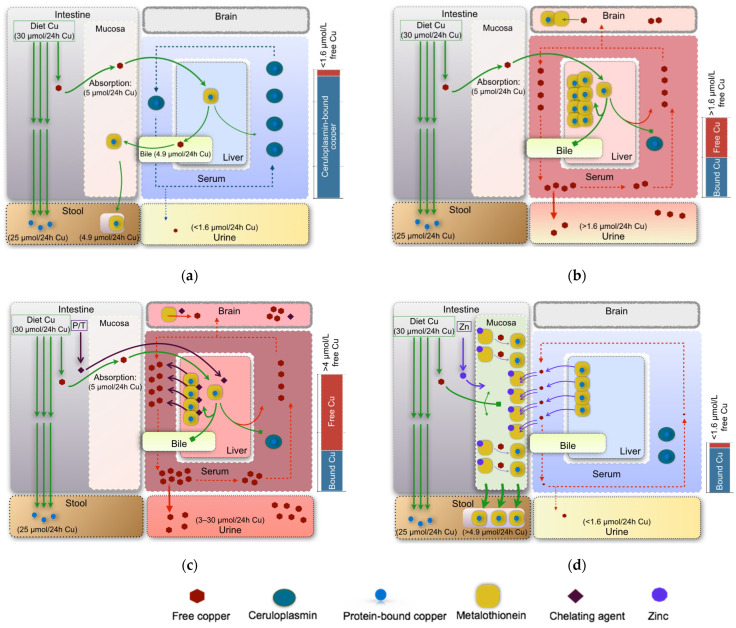
Schematic copper balance: Panel (**a**) reflects normal copper metabolism and a normal copper (Cu) balance in which the daily amount of copper ingested from the diet (equal to approximately 30 µmol or 2 mg per 24 h [24 h] [72,73]). Combined excretion of copper via feces and urine is about 25–30 µmol equal to 1.5–2 mg/24 h. Almost 85–95% of total serum copper is tightly bound to ceruloplasmin (shown with a sea green oval) [74], and the rest is in unbound or loosely bound (free) form (shown with a dark red hexagon). Panel (**b**) reflects impaired copper metabolism in Wilson’s disease (WD), in conditions in which the synthesis of holoceruloplasmin and the excretion of free copper with bile are diminished. Without treatment, the total excretion of copper via the urine and stools remains slightly lower than normal [37,61,75], thereby favoring copper toxicity. The serum free copper can be bound by metallothioneins (shown with yellow squares), deposited in organs (mainly liver), or excreted in the urine. Panel (**c**) illustrates WD treatments based on chelating agents (i.e., penicillamine (P), trientine (T)), which stimulate the transfer of copper from its organ deposits into the blood, which can induce free copper intoxication. The kidneys counteract high free copper levels by increasing urinary excretion. Panel (**d**) depicts WD treatment with zinc (Zn), which stimulates the synthesis of intestinal metallothioneins. Since metallothioneins have a greater affinity for copper compared to zinc, the treatment antagonizes exogenous copper absorption (shown with a small green tetragon indicating an impaired entrance of copper from the intestine into serum) and may also neutralize free copper in serum (shown with a lower amount of free copper and light bluish serum color and also by double purple arrows). Intestinal metallothionein-bound copper is excreted in the feces when mucosal cells are sloughed. Overall, the copper balance becomes negative, and the total copper content, including the accumulated organ deposits, is gradually reduced (shown with lower yellow squares in the liver and bluish serum compared to that in Figure 1b). Bound copper is shown in blue and free copper is shown in red. Small green tetragons indicate impaired/blocked pathways. A bluish serum indicates a normal serum free copper and a reddish serum indicates an increased serum free copper level. Similarly, a bluish liver indicates normal serum free copper levels while a reddish liver indicates increased serum free copper levels.

**Table 1 ijms-23-09316-t001:** Relative frequency of diagnostic markers present in a retrospective study of Wilson’s disease (WD) patients [94,95].

Criteria	Hepatic and Neurologic WD [94]	Hepatic WD [95]	Neurologic WD [95]
Serum ceruloplasmin (<200 mg/L)	88%	59%	85%
Urine copper (>1.6 μmol/24 h)	87%	90%	78%
Liver copper content (>4 μmol/g)		90%	93%
Histological signs of liver damage	73%		
Kayser–Fleischer rings (with slit lamp examination)	66%	41%	90%
Mutations in *ATP7B* on 1 or 2 alleles	85%		
Non-ceruloplasmin bound copper (>4 μmol/L)	86%		

Normal values: Serum free copper: 1.6–2.4 μmol/L (equal to 10–15 µg/dL), urine copper: 0.3–0.8 μmol/24 h (equal to 20–50 µg/24 h or <75 µg/g creatinine), serum copper: 11–24 μmol/L (equal to 70–150 µg/dL), hepatic copper: 0.3–0.8 μmol/g dry weight tissue (equal to 20–50 µg/g), ceruloplasmin: 180–350 mg/L (equal to 18–35 mg/dL), aspartate aminotransferase (AST): 10–35 U/L, alanine aminotransferase (ALT): 10–40 U/L.

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
