# Peer review of "The Role of Zinc in the Treatment of Wilson’s Disease"

_ijms, 2022, doi:10.3390/ijms23169316_

Round 1

Reviewer 1 Report

I described comments in the below file.

Author Response

Reviewer 1 comments and authors’ responses:

This review paper reports that pathology, mechanism of copper toxicity and treatment of Wilson disease, focusing the role of zinc in the treatment of Wilson disease. I think this paper is valuable to understand Wilson disease and treatment with zinc.

Response: We are grateful for the reviewer’s positive impression of our review of the role of zinc therapy in Wilson’s disease and for their constructive comments.

Some comments are in below.

1)     Figure1: I hope that papers cited values in Fig.1 (a), for example, Diet Cu 30 µmol/d Cu, will be referred.

Response: The amount of copper ingested in food in patients with Wilson’s disease is from the same as that of normal subjects and is estimated at approximately 30 umol/24h (approximately 1.5 to 2 mg per day (Reference 72 and 73). Combined excretion of copper via feces and urine is about 25–30 µmol equal to 1.5–2 mg per day (24h), and slightly lower in Wilson’s disease patients (References 37, 61, and 75). We added these references to the caption.

The unit of urinary copper in (a) and (b) is μmol/d, while that in (c) and (d) is µmol/L. Is it right?

Response: We corrected all the referenced units to μmol/24h.

 Figure 1 (b): The liver damage of Wilson disease seems to be caused by free-copper accumulation in the liver, resulting in oxidative stress in the cells.  At that time, accumulated free copper in the hepatocytes release into blood (Nagral A et al. J Clin Expe Hepatol 2019; 9: 74-88; Kodama H et al. Current Drug Metabolism 2012; 13: 237-350). However, figure 1 (b) does not show accumulated free copper in the liver.

Response: Free copper needs to be bound to a protein in order to accumulate in an organ. Accumulated copper has been shown in figure 1b as copper (shown in blue) bound to metallothionein (shown in yellow) and also described in the caption. We tried to further clarify this in the caption and have added a legend.

Figure 1 (d): As the authors describes in lines 212 and 213, zinc salts act not only to reduce copper absorption, but also to induce metallothionein in hepatocytes, resulting in a de-coppering of organs.  I hope that the above act of zinc is shown in Figure 1 (d)

Response: The de-coppering action of zinc occurs mainly by the induction of intestinal metallothionein (25-fold increase) more than its induction in the liver. Due to the high affinity of dietary and serum copper to intestinal metallothionein, its entrance into serum is blocked, while it is excreted via sweat and secretion from the serum into the intestine to bind into intestinal metallothionein; these are shown in figure 1 (d). The latter is indicated by the double purple arrows. We have further clarified the points in the caption and added a legend to improve reader interpretation.

Figure 1 (d): What is the small green tetragon in mucosa of figure 1 (d)?

Response: The green tetragon in mucosa of figure 1d indicates the impaired entrance of copper from intestine into serum. We added an explanation to the caption to clarify this point and added a legend.

2)     Table 1: Please correct the unit of liver copper content to µmol/g dry weight.

 Response: We corrected the unit in table 1.

3)     Line 333 and 334: Author describe that zinc therapy is effective in fulminant hepatic failure, but only few patients (refs. 130,131). I am afraid that zinc therapy seems to be recommended for fulminant hepatic failure also.  As described in some Guidelines, including ref 4, and 75, it should be emphasized that the first choice treatment of fulminant hepatic failure is liver transportation.

Response: We have revised the text to emphasize the current recommendations and to propose the possibility of trying zinc to decrease the free copper levels, rather than using chelators which increase free copper level possibly causing or worsening hemolytic anemia. Zinc is safe and it may ameliorate toxic levels of free copper before a transplantation is possible. Alternatively, plasmapheresis can be used before the treatment with zinc salts to rapidly reduce free copper levels prior to the start of zinc treatment, thereby increasing the likelihood of zinc maintaining a reduced free copper level. Our positive experience on initiating zinc in an acute phase of liver failure [Reference 136] in addition to comparable reports in the literature [References 134 and 135] support the proposal, which demands further investigation. The limited number of patients treated with zinc in an acute liver failure phase might be due to current recommendations that hinder using zinc as an initial treatment option.

4)     The authors point that zinc is an equally effective as chelators. I agree the points in case of maintenance therapy, asymptomatic patients and patients with mild hepatic damages. However, chelating agents are recommended as initial therapy in patients with moderate and severe hepatic disturbances to remove promptly accumulated copper in the body (refs 4 and 75). Please show that zinc is an equally effective as chelators in initial therapy for patients with moderate and severe hepatic damages if that is true.

Response: The equality of the efficacy of zinc and chelators is based on the results of two recent systematic reviews and meta-analyses [References number 102 and 103]. The recommendation of chelators in patients with moderate and severe hepatic disturbances is based on experts’ opinions and not based on firm head-to-head comparison of different options. There are reports of the efficacy of initial zinc therapy in patients with moderate and severe hepatic damages: References 83 and 134–136. Neither will prove or disprove the superiority of either option. Therefore, it is worthwhile to begin treatment with zinc as it is safe, effective, and affordable, and then follow the disease progression or control and reassess treatment options.

Reviewer 2 Report

This is a well written summary of the role of zinc in the treatment of Wilson’s disease. the authors conclusion that Zinc is as effective as other anticopper agents due to its low level of toxicity is a valid one. however, as noted, due to the lack of consensus between the different jurisdictions (as highlighted in the variances between CPGs), lack of RCTs and great variance between patients presenting with WD, it will be challenging to convince some clinicians on the use of Zinc as first line therapy, especially for hepatic presentations.

The authors have done well to present the data and safety considerations in section 6 “Effectiveness and safety profile of treatment options”, however, due to the potential for treatment failure (especially in hepatic cases) and the need for close monitoring especially for those without both a normal serum ALT and appropriate urine copper excretion, the authors should further highlight the potential disadvantages of using Zinc.

Author Response

Reviewer 2 comments and authors’ responses:

This is a well written summary of the role of zinc in the treatment of Wilson’s disease. the authors conclusion that Zinc is as effective as other anticopper agents due to its low level of toxicity is a valid one. however, as noted, due to the lack of consensus between the different jurisdictions (as highlighted in the variances between CPGs), lack of RCTs and great variance between patients presenting with WD, it will be challenging to convince some clinicians on the use of Zinc as first line therapy, especially for hepatic presentations.

Response: We thank the reviewer for such a positive impression and for their constructive comments. We do agree about the challenge clinicians may face, either due to lack of experience or awareness of zinc treatment effectiveness or due to the imposition of a sequential trial of penicillamine and trientine before zinc in order to satisfy drug insurance company regulations (in the US). Despite the lack of randomized controlled trials to provide data for a more informed comparison of which treatment performs better, our review aims to encourage clinicians to reconsider the basis of different approaches to the treatment of Wilson’s disease and rethink the efficacy and risk/safety of the proposed option.

The authors have done well to present the data and safety considerations in section 6 “Effectiveness and safety profile of treatment options”, however, due to the potential for treatment failure (especially in hepatic cases) and the need for close monitoring especially for those without both a normal serum ALT and appropriate urine copper excretion, the authors should further highlight the potential disadvantages of using Zinc.

Response: We are grateful to the reviewer for the compliment. We highlighted the potential challenges of using zinc therapy and the need for careful monitoring of its efficacy and patient adherence to the treatment, as is the case for all treatment options. A major disadvantage of zinc is the inability of the pharmaceutical companies to generate enormous profit at the expense of and risk to patients (References 84 and 124)!

Round 2

Reviewer 1 Report

The revised paper made a modification adequately  according to the comments.